# SASP-Dependent Interactions between Senescent Cells and Platelets Modulate Migration and Invasion of Cancer Cells

**DOI:** 10.3390/ijms20215292

**Published:** 2019-10-24

**Authors:** Claudio A. Valenzuela, Ricardo Quintanilla, Alexandra Olate-Briones, Whitney Venturini, Daniel Mancilla, Angel Cayo, Rodrigo Moore-Carrasco, Nelson E. Brown

**Affiliations:** 1Center for Medical Research, Medical School, University of Talca, Talca 3460000, Chile; cvalenzuela@utalca.cl (C.A.V.); r_mquintanilla@hotmail.com (R.Q.); amolateb@gmail.com (A.O.-B.); whitneyventurini@gmail.com (W.V.); daniel.enrique.mancilla@gmail.com (D.M.); acayo@utalca.cl (A.C.); 2Núcleo Científico Multidisciplinario, Universidad de Talca, Talca 3460000, Chile; 3Faculty of Health Sciences, University of Talca, Talca 3460000, Chile; 4Programa de Investigación Asociativa en Cáncer Gástrico (PIA-CG), Talca 3460000, Chile

**Keywords:** cellular senescence, Palbociclib, SASP, platelets

## Abstract

Alterations in platelet aggregation are common in aging individuals and in the context of age-related pathologies such as cancer. So far, however, the effects of senescent cells on platelets have not been explored. In addition to serving as a barrier to tumor progression, cellular senescence can contribute to remodeling tissue microenvironments through the capacity of senescent cells to synthesize and secrete a plethora of bioactive factors, a feature referred to as the senescence-associated secretory phenotype (SASP). As senescent cells accumulate in aging tissues, sites of tissue injury, or in response to drugs, SASP factors may contribute to increase platelet activity and, through this mechanism, generate a microenvironment that facilitates cancer progression. Using in vitro models of drug-induced senescence, in which cellular senescence was induced following exposure of mammary epithelial cells (MCF-10A and MCF-7) and gastric cancer cells (AGS) to the CDK4/6 inhibitor Palbociclib, we show that senescent mammary and gastric cells display unique expression profiles of selected SASP factors, most of them being downregulated at the RNA level in senescent AGS cells. In addition, we observed cell-type specific differences in the levels of secreted factors, including IL-1β, in media conditioned by senescent cells. Interestingly, only media conditioned by senescent MCF-10A and MCF-7 cells were able to enhance platelet aggregation, although all three types of senescent cells were able to attract platelets in vitro. Nevertheless, the effects of factors secreted by senescent cells and platelets on the migration and invasion of non-senescent cells are complex. Overall, platelets have prominent effects on migration, while factors secreted by senescent cells tend to promote invasion. These differential responses likely reflect differences in the specific arrays of secreted senescence-associated factors, specific factors released by platelets upon activation, and the susceptibility of target cells to respond to these agents.

## 1. Introduction

Cellular senescence is a unique form of irreversible cell cycle arrest characterized by specific changes in gene expression, morphology, and function [1,2]. First described in primary cells subjected to long-term culture in vitro [3], a similar phenotype can be triggered prematurely in response to various forms of stress, including genotoxic, oxidative, oncogenic, and therapeutic stress [4]. Distinguishing features of senescent cells are a large and flat morphology, the presence of numerous vacuoles in the cytoplasm, the formation of heterochromatic foci in the nucleus, and the upregulation of senescence-associated β-galactosidase (SA-β-Gal) activity [5,6,7]. In addition, proliferative arrest in senescent cells is associated with activation of the p53–p21^KIP1/CIP1^ and/or p16^INK4A^–pRB tumor suppressor pathways [8,9]. As these pathways are commonly disrupted in cancer cells, cellular senescence has traditionally been considered a potent failsafe program that prevents the development of cancer by limiting aberrant proliferation of preneoplastic cells [10].

Besides these in vitro features of senescent cells, several lines of evidence point to the existence of functionally distinct types of senescence in vivo. Thus, a self-limited form of cellular senescence has been described in vivo in the context of embryonic development and wound healing [11,12,13,14,15]. In contrast, a chronic or persistent form of senescence, characterized by the accumulation of senescent cells, is thought to contribute to aging, chronic inflammation, and cancer [16,17,18]. That persistent senescent cells contribute to organismal aging and the emergence and progression of age-related diseases [11,12,19] has recently been demonstrated in vivo. For example, the elimination of senescent cells in aging mice increases the health span and reduces the consequences of age-related pathologies or chemotherapy [20,21].

Although cellular senescence has been linked to age-dependent stem or progenitor cells exhaustion [20,22,23], the sole inability of senescent cells to divide cannot explain all the in vivo consequences of cellular senescence. Instead, its long-term deleterious effects likely depend on the capacity of senescent cells to produce and secrete a variety of soluble and insoluble factors, a feature referred to as the senescence-associated secretory phenotype (SASP) [16]. Interestingly, pro-inflammatory mediators (e.g., chemokines and cytokines) are among the most highly conserved SASP components across cell types and senescence-inducing stimuli [24,25,26,27,28]. It has been speculated that the accumulation of senescent cells might contribute to the low-grade, chronic, and systemic inflammation associated with aging and aging phenotypes [29]. In this context, the presence of senescent cells in tissues may promote the acquisition of neoplastic features in adjacent cells or otherwise foster the generation of a proinflammatory environment that facilitates tumorigenesis [26,28,30,31,32,33,34,35].

Similar to senescent cells, platelets also play an active role in age-related conditions, most typically atherosclerosis and thrombosis, but also inflammation and cancer [36]. Indeed, accumulating evidence indicates that platelets participate in the orchestration of numerous inflammatory processes and are involved in almost every step of tumorigenesis [37,38,39,40]. Particularly, complex bi-directional interactions between cancer cells and platelets have been described [41,42]. For example, tumor cells are capable of inducing platelet aggregation through their ability to produce and secrete platelet-activating factors, including thrombin, ADP, thromboxane A2, metalloproteinases, and tissue factor [43,44,45]. Conversely, platelets can also affect tumor growth, angiogenesis, and metastasis through the release of several growth factors (PDGF-A/B, IGF, TGF-beta1, VEGF, SDF-1, among others) and/or by direct contact with tumor cells [46,47,48]. Among platelet-released growth factors, VEGF, an important proangiogenic factor, is thought to modulate tumor angiogenesis [49,50]. Indeed, platelets derived from cancer patients present higher levels of VEGF in their granules than platelets from healthy individuals [51]. Finally, the role of platelets during metastasis has been amply documented, a function that results, at least in part, from the ability of platelets to induce epithelial-to-mesenchymal transition (EMT) [52] and to favor the formation of primary metastatic niches [53]. Additionally, platelets protect circulating tumor cells from shear damage and immune-mediated tumor surveillance [53,54,55] and further assist the extravasation of tumor cells at sites of metastasis, thus functioning as a bridge between endothelial and tumor cells [56,57,58].

On the basis of these observations, we hypothesized that platelets and senescent cells may act synergistically in chronic inflammatory processes and tumor progression. However, the potential role of senescent cells and their secreted factors as modulators of platelet function is largely based on indirect evidence [37]. For example, a decrease in bleeding time (a surrogate for platelet activity) and an elevation of markers of platelet activation have both been correlated with physiological aging [59]. Moreover, platelets from older individuals display an increased aggregation response to ADP and collagen compared to those from younger individuals [60]. Interestingly, interleukin 6 (IL-6), a proinflammatory cytokine and one of the most prominent factors secreted by senescent cells [61], can directly activate platelets [62].

Herein, we show that factors secreted by Palbociclib-induced senescent cells are able to enhance platelet aggregation and likely mediate the recruitment of platelets to sites of cellular senescence in vitro. Nevertheless, the effects of senescent cells and platelets on migration and invasion of non-senescent cells are complex. Overall, platelets seem to have a prominent role in migration, while factors secreted by senescent cells tend to promote invasion. Therefore, the differential responses to factors secreted by senescent cells and platelets are dependent on the specific array of senescence-associated factors, factors released by platelets upon activation, and the susceptibility of the target cells to respond to these inputs.

## 2. Results

### 2.1. Analyses of Selected SASP Factors in Senescent AGS, MCF-10A, and MCF-7Cells

In order to examine the effects of SASP factors on platelet aggregation, we first optimized the experimental conditions for drug-induced senescence in three cellular systems, namely, human gastric cancer cells (AGS), immortalized mammary epithelial cells (MCF-10A), and breast cancer cells (MCF-7). For this, both the minimal concentration and the time required to induce senescence using Palbociclib, a CDK4/6 inhibitor that induces senescence as a main outcome [63,64], were determined (data not shown). Cellular senescence could be induced in all three cell lines following exposure to Palbociclib for 96 h (Appendix A). Thus, upon Palbociclib treatment, AGS, MCF-10A, and MCF-7 cells displayed increased levels of SA-β-Gal activity, a widely used assay for senescence detection (Appendix A) [65].

In order to assess the expression of SASP factors in Palbociclib-induced senescent cells, we selected 16 factors on the basis of their previously documented capacity to directly or indirectly affect platelet aggregation or the fibrinolytic system. As shown in Figure 1A–C, qRT-PCR-based expression profiles of senescent AGS, MCF-10A, and MCF-7 cells differed widely, likely reflecting differences in cell lineage, degrees of neoplastic transformation, or both. Interestingly, and contrary to what was expected, most factors were downregulated in senescent AGS cells when compared to their non-senescent counterparts (Figure 1A). This was in stark contrast to the overall upregulation of gene expression observed in senescent MCF-10A (Figure 1B) and MCF-7 (Figure 1C) cells. Factors upregulated in all three senescent cells included *IL1-β*, *G-CSF*, and *ICAM-1*. Particularly prominent was the expression of *PAI-2* in senescent MCF-10A and MCF-7 cells, which represent non-tumorigenic and tumorigenic mammary epithelial cells, respectively. Thus, the expression of *PAI-2*, an important regulator of the fibrinolytic system, was increased about four-fold in senescent MCF-10A and MCF-7 cells compared to DMSO-treated control cells (Figure 1B,C). Of note, a trend that was similar to that in the mRNA-based expression profiles was observed when factors secreted by senescent cells were assessed by cytokine arrays. Thus, an overall reduced production of factors secreted by senescent AGS cells compared with senescent MCF-10A cells was evident (Appendix A).

Among the pro-inflammatory factors that were upregulated at the mRNA level in all three cell lines analyzed, IL-1β is particularly interesting as it has been related to an increase in platelet activation and adhesion in the context of diverse inflammatory processes [66,67]. Of note, following ELISA analyses, we only observed a detectable increase in the levels of IL-1β secreted into media for senescent MCF-10A cells (Appendix A), despite the upregulation of *IL-1β* mRNA in senescent cells of all three cell lines (Figure 1A–C). These experiments highlight the relevance of determining the protein levels of SASP factors.

### 2.2. The Effect of Factors Secreted by Senescent Cells on Platelet Aggregation

While several reports have examined the effects of senescent cells and their secreted factors in different tissue compartments [22,68,69,70], the roles of senescent cells and their secreted factors as modulators of platelet activity have not been explored [37]. To determine the effects of SASP factors on platelet aggregation, platelets were exposed to conditioned medium derived from senescent and non-senescent cells under aggregating conditions. As shown in Figure 2, the amplitude of the aggregation curves, indicative of total platelet aggregation, increased by 18.3% when platelets were incubated with conditioned media derived from senescent MCF-10A cells in comparison to platelets that were exposed to media conditioned by non-senescent cells (Figure 2A,B, middle panels). Likewise, an increase in aggregation was observed when platelets were exposed to conditioned media derived from senescent MCF-7 cells (Figure 2A,B, right panels; 12.8% increase of aggregation). Despite a similar trend, the values of platelet aggregation following the exposure of platelets to media conditioned by senescent and non-senescent AGS cells were not statistically significant (Figure 2A,B, left panels).

### 2.3. Adhesion Assays of Senescent Cells and Platelets

We also speculated that, prior or concomitant to aggregation, the paracrine actions of senescent cells might involve the recruitment of platelets to sites of cellular senescence. In order to explore this possibility in vitro, senescent and non-senescent MCF-10A, AGS, and MCF-7 cells were incubated with platelets that had been previously labeled with Calcein-AM (green fluorescence). As shown in Figure 3, increased accumulation of platelets around all three types of senescent cells was observed, suggesting attraction of platelets to sites of senescence.

### 2.4. Effect of SASP Factors and Platelets on Migration and Invasion of Non-Senescent Cells

It has been speculated that the chronic release of inflammatory molecules or other factors by persistent senescent cells might contribute to cancer progression [4]. Accordingly, factors secreted by senescent fibroblasts can promote proliferation and migration of premalignant epithelial cells [26,28,31,32]. Mirroring these effects, platelets can also enhance various aspects of cancer progression [71]. Interestingly, even a transient interaction with platelets is sufficient to induce EMT of tumor cells, enhancing their migrating and invasive capabilities [72]. On the basis of these observations and the evidence that factors secreted by senescent cells can enhance platelet aggregation (see Figure 2), we set out to explore the individual and combined effects of factors secreted by senescent cells and platelets on migration and invasion of non-senescent cells using in vitro wound-healing and Transwell assays. The results showed that the sole addition of platelets led to an increase in the migratory capabilities of non-senescent MCF-10A and MCF-7 cells (Figure 4A,B, middle and lower panels). While a tendency to increase migration of AGS cells was also observed in the presence of platelets, this increase was not statistically significant (Figure 4A,B, upper panels). Surprisingly, the exposure of non-senescent cells solely to media conditioned by senescent cells led to an increase in the migration of MCF-7 cells (Figure 4A,B, lower panels), but it was inconsequential for MCF-10A cells or even led to a slight reduction in the ability of non-senescent AGS cells to migrate (Figure 4A,B, middle and upper panels). In these experiments, platelets seemed to exert a dominant effect over factors secreted by senescent cells in their ability to enhance migration, as evidenced by the effects of the combined treatments (Figure 4A,B).

In contrast to migration, media conditioned by senescent AGS, MCF-10A, and MCF-7 cells increased invasion of the respective non-senescent cells in the absence of platelets (Figure 5A,B). Of note, while invasion of non-senescent AGS cells increased in the presence of platelets (Figure 5A,B, upper panels), a similar response was not seen in non-senescent mammary cells exposed to platelets alone (Figure 5A,B, middle and lower panels). On the other hand, exposure to both platelets and senescent conditioned media led to a significant enhancement (compared to the invasion effects of individual treatments) in the invasive capabilities of non-senescent MCF-7 cells (Figure 5A,B, lower panels), but not in those of AGS and MCF-10A cells (Figure 5A,B, upper and middle panels). Altogether, these results indicate that factors secreted by senescent cells and platelets produce complex outcomes in the ability of target, non-senescent cells, to migrate and invade. These complexities might be related to differences in the array of factors secreted by both senescent cells and platelets and/or to the specific features of the target cells receiving these signals.

## 3. Discussion

Cellular senescence has been traditionally described as a non-proliferative process that limits aberrant proliferation of preneoplastic cells [10]. However, far from being passive bystanders, senescent cells are known to be able to alter tissue microenvironments through their capacity to synthesize and secrete a wide range of bioactive molecules, a feature known as the SASP [73]. Several lines of evidence indicate that the chronic release of inflammatory molecules and other soluble factors by persistent senescent cells contribute to aging and the development of age-related pathologies [4]. Importantly, factors secreted by senescent cells can also have tumor-promoting effects. For example, factors secreted by senescent fibroblasts can promote proliferation and migration of premalignant epithelial cells [26,28,32,34] or induce tumor vascularization and invasiveness [31]. Interestingly, part of these effects might reflect the ability of SASP factors to stimulate the acquisition of stem cell properties by neighbor cells. [33,35]. Similarly, although the induction of senescence may constitute a desirable immediate outcome of cancer therapy, senescent cells that fail to be eliminated by the immune system likely contribute to a process of tissue remodeling that favors cancer recurrence or premature aging [74]. This is particularly relevant when using cancer drugs that induce senescence as their main mechanism of action, including the newly developed cyclin-dependent kinase inhibitors [75].

While several reports have explored the effects of senescent cells and their secreted factors in different tissue compartments [23,68,69,70], the role of senescent cells and their secreted factors as modulators of platelet activity has not been explored [40]. Similar to senescent cells, platelets play an active role in age-related conditions, most typically atherosclerosis and thrombosis, but also inflammation and cancer [36]. To study the functional interactions between senescent cells and platelets, an in vitro model of drug-induced senescence was first implemented. This model was largely based on the use of Palbociclib, an inhibitor of cyclin-dependent kinases 4 and 6 (CDK4/6), important regulators of cell cycle progression. Of note, the chief mechanism accounting for the efficacy of CDK4/6 inhibitors as anti-cancer drugs is cellular senescence [64].

After confirming Palbociclib-induced senescence in AGS, MCF-10A, and MCF-7 cells, platelet aggregation assays were carried out in which purified platelets were exposed to the media conditioned by senescent and non-senescent cells. On the basis of the results presented in Figure 2, we conclude that factors secreted by senescent cells enhance platelet aggregation. Differences in the extent of platelet aggregation produced by media conditioned by different senescent cells were likely due to differences in their SASPs, which in turn reflected differences in cell lineage or degree of transformation. Thus, when the expression of a subset of factors secreted by senescent cells—which were selected on the basis of their ability to alter platelet activity (Appendix A)—was assessed, significant differences in the expression profiles of senescent AGS, MCF-10A, and MCF-7 cells were observed (Figure 1A–C). Of note, most of the factors examined were negatively regulated in senescent AGS cells (Figure 1A). This was in stark contrast with the overall overexpression of most of the genes examined in senescent MCF-10A and MCF-7 cells (Figure 1B,C).

Although far from exhaustive, our expression analyses highlight the potential contribution of factors secreted by senescent cells in the modulation of platelet activity. For example, Schroder et al. (2019) have shown that *PAI-2* (*SERPINB2*), a factor highly expressed in senescent MCF-10A and MCF-7 cells, has a central role in hemostasis. Thus, *PAI-2*-deficient mice have shorter bleeding times than wild-type mice, without changes in platelet count and volume [76]. In addition, these animals have shorter reaction times (onset of coagulation) and tend to have a greater and faster secretion of ATP stimulated by collagen and arachidonic acid [76]. On the other hand, the inhibitor of macrophage migration (MIF), a growth factor also produced and secreted by platelets, was elevated in the conditioned medium derived from both senescent MCF-10A and AGS cells. MIF has a central role in the regulation of inflammation and thrombosis, functions that would be, at least partially, related to its ability to interact with Gremlin-1, another important inflammatory and thrombotic regulator in platelets [77]. Likewise, an increase in the secretion of IL-8 and Gro-α was evident only in the media conditioned by senescent MCF-10A cells. All these factors are involved in inflammation-associated chemotaxis [78,79].

A more sensitive assay revealed the presence of elevated levels of the IL-1β in media conditioned by senescent MCF-10A cells but not in media conditioned by senescent AGS or MCF-7 cells (Figure 1C), despite the upregulation of IL-1β at the mRNA level in all three senescent cell lines (Figure 1A,B). Interestingly, platelets express the IL-1β receptor (IL-1R1) and therefore respond to IL-1β, establishing complex autocrine and paracrine signaling loops that might promote the adhesion of platelets to endothelial cells during inflammatory processes [66,67]. Taken together, differences in the expression of selected SASP components support the existence of different senescence-associated secretory profiles depending on cell lineage and degree of neoplastic transformation. These differing secretory profiles, in turn, seem to contain different platelet-aggregating properties.

We observed that platelets have the capacity to adhere to senescent cells in vitro, further suggesting the existence of paracrine interactions between senescent cells and platelets. It is therefore tempting to speculate that senescent cells, through the SASP, could recruit platelets to sites of senescence-induced inflammation in vivo. Given the diversity of molecules secreted by senescent cells and platelets, their effects on tissue microenvironments are expected to be complex, likely contributing to endothelial dysfunction [26], as well as proliferation [80], migration, and invasion [81] of precancerous cells. We therefore explored the individual and combined effects of factors secreted by senescent cells and platelets on migration and invasion of non-senescent cells. Altogether, our results indicate that senescent cells and platelets produce complex outcomes in the ability of non-senescent cells to migrate and invade. For example, platelets alone seem to increase the migration of non-senescent MCF-10A and MCF-7 cells. However, an increase in migration upon exposure to senescent cells-conditioned medium was only evident in MCF-7 cells. While no enhancing migratory effects were observed with the combined treatment, the effects of platelets seem to dominate over the effects of senescent cells-conditioned media in these assays. In contrast to migration, factors secreted by all three senescent cells seemed to increase the invasion of non-senescent cells. Interestingly, platelets alone seemed to also increase the invasion of non-senescent AGS cells.

In summary, we describe the existence of Palbociclib-induced SASP profiles with different platelet-aggregating capacities depending on cell lineage and degree of neoplastic transformation. Surprisingly, platelets and media conditioned by senescent cells had diverging effects on in vitro cell migration and invasion.

## 4. Materials and Methods

### 4.1. Cell Culture

Human AGS cells (ATCC code: CRL-1739) were kindly provided by Dr. Alejandro Corvalan (Pontifical Catholic University of Chile, Santiago, Chile). Similarly, MCF-10A (ATCC code: CRL-1031) and MCF-7 cells (ATCC code: HTB-22) were provided by Dr. Philip W. Hinds (Tufts University School of Medicine, Boston, MA, USA). AGS cells were maintained in RPMI-1640 (1:1) medium (HyClone, Thermo Scientific, Waltham, MA, USA) supplemented with 10% FBS (HyClone, Thermo Scientific). MCF-10A cells were maintained in complete MEGM medium (Lonza Group AG, Basel, Switzerland), and MCF-7 cells were maintained in high-glucose DMEM, supplemented with 10 μg/mL of insulin. All media were supplemented with 25 mg/mL of Gentamicin and 250 ng/mL Amphotericin B (Sigma, St. Louis, MO, USA).

### 4.2. Senescence Induction and Generation of Conditioned Media

AGS, MCF-10A, and MCF-7 cells were induced to senesce by treatment with Palbociclib (PD033299, Pfizer, New York, NY, USA), a specific inhibitor of cyclin D–CDK4/6 kinase complexes. For this, 4 × 10^5^ cells, seeded on 10 cm-diameter plates, were treated with 1 μΜ of Palbociclib for 4 days. After this time, the cells were harvested for total RNA and protein extraction. Detection of cellular senescence, using senescence-associated (SA)-β-Galactosidase activity, was carried out as described before [65]. Briefly, cells cultured on coverslips contained in 6-well plates (5 × 10^4^ cells per well) were treated with Palbociclib or vehicle (DMSO), fixed with 2% paraformaldehyde/0.2% glutaraldehyde in PBS, and incubated at 37 °C overnight in staining solution (40 mM citric acid/sodium phosphate pH 6.0, 5 mM of potassium hexacyanoferrate (II) tri-hydrate, 5 mM potassium hexacyanoferrate (III), 150 mM sodium chloride, 2 mM magnesium chloride, 50 mg/mL X-Gal). The proportion of senescent cells (blue cells) was determined by microscopic inspection (40× magnification, BX53 microscope, Olympus, Hongkong, China) of at least three independent experiments. For each experiment, the relative proportion of positive cells was determined by analyzing the colored area in Palbociclib- and DMSO-treated cells using ImageJ program and considering 5 separate microscopic fields.

Culture media, conditioned by Palbociclib- or vehicle (DMSO)-treated cells, were collected, filtered, and used immediately in platelet aggregation and migration/invasion assays. In order to minimize the effects of serum, the cells were cultured in a minimal volume of serum-free medium for 24 h before harvesting.

### 4.3. Platelet Aggregation Assays

Platelets were obtained from blood samples (20 mL) obtained from healthy volunteers. Healthy volunteers were subjected to venous puncture and blood withdrawal only after signing an informed consent document. Before the procedure, a short survey was applied in order to ensure that the individuals met the inclusion criteria. Both the informed consent and the survey had been previously approved by the Scientific Ethics Committee of the University of Talca (code number 2015-102-NB, approved on 15 April 2015). The following were the inclusion criteria used to select blood donors: male gender, age between 18 and 25 years, absence of chronic or infectious diseases, absence of hematological or coagulation diseases, body mass index (BMI) values between 18.5 and 24.9, cessation of consumption of nonsteroidal anti-inflammatory drugs at least 10 days before blood withdrawal, and no alcohol and tobacco consumption during 10 days before blood withdrawal. Platelet-enriched plasma and washed platelets were obtained as described before [82]. Briefly, blood samples were collected in tubes containing 3.2% of the anticoagulant sodium citrate, in a 9:1 ratio (blood/anticoagulant) and centrifuged at 240 g for 10 min at 4 °C to obtain platelet-rich plasma (PRP). Two-thirds of the PRP was removed and again centrifuged at 650 g for 10 min at 4 °C. The pellet was then washed with HEPES-Tyrode buffer containing prostaglandin E1 (PGE1, 120 nmol/L, pH 6.2). Finally, the washed platelets were re-suspended in HEPES-Tyrode buffer at a concentration of 2 × 10^5^ platelets/μL. The number of platelets was quantified using an automated hematology analyzer (BC-2800 Mindray Analyzer, Shenzhen, China).

For platelet aggregation assays, 480 μL of washed platelets were combined with culture medium previously conditioned by senescent or non-senescent cells (~50 μL), and the mixture was incubated for 5 min at 37 °C, under constant stirring. Platelet aggregation was measured in the presence of aggregating concentrations of ADP (4.0 μM) and fibrinogen (275 μg/mL; Cayman Chemical, Ann Arbor, MI, USA; catalog number 16088). Transmittance was determined using an aggregometer in real time for a 6 min period. All measurements were performed in triplicate and analyzed by the AGGRO/LINK software (Chrono-Log, Havertown, PA, USA).

### 4.4. Quantitative Real-Time PCR (qRT-PCR)

To determine the expression of factors secreted by senescent cells, the SYBR Green/ROX-based quantitative PCR Master Mix was used (Life Technologies, Carlsbad, CA, USA). All qRT-PCR reactions (20 μL each) were run on a Mx3000P PCR detection system (Stratagene, Cedar Creek, TX, USA) using the following cycling conditions: initial denaturation at 95 °C for 10 min, 40 cycles at 95 °C for 15 s, 60 °C for 15 s, and 72 °C for 20 s. Final primer concentrations were 250 nM for all reactions. Complementary DNAs (cDNAs) were synthesized from 1 μg of DNase-treated total RNA using a first-strand cDNA synthesis kit (Fermentas, Life Sciences, Waltham, MA, USA) according to the manufacturer’s instructions. The product of the first-strand reaction was diluted tenfold; 2 μL of this dilution were used for each qRT-PCR reaction. The cycle threshold (Ct) was determined manually as the point where the R^2^ value for the standard curve reached its highest point [83]. qRT-PCR reactions were run in duplicate, and the values for each sample were plotted as mean ± SD of three biological replicates. The efficacies of all primers were previously evaluated, and only those giving efficiencies of 90–100% were selected. Additionally, agarose gel electrophoresis and melting curve analyses were performed in order to confirm the specificity of the PCR products. The expression of each gene was normalized to that of the ribosomal protein L19 (RPL19). All RT-qPCR primers are listed in Appendix A.

### 4.5. ELISA

The levels of the pro-inflammatory cytokine IL-1β in conditioned media were determined with an ELISA kit (Bender MedSystems, GmbH, Vienna, Austria) according to the manufacturer’s instructions (IL-1β ELISA kit, High sensitivity, cat. No. BMS224HS). Briefly, 50 μL of conditioned media (CM) was added to wells containing an immobilized anti-IL-1β antibody. Biotin-streptavidin and colorimetric reagents were added for signal amplification. Signals were detected by a Synergy HTX Multi-Mode Microplate reader (Biotek instrument, Winooski, VT, USA) at 450 nm. The density values for each sample were determined using the standard curve of human IL-1β (10–0.16 pg/mL) to calculate the real values.

### 4.6. Cytokine Arrays

In order to identify soluble factors secreted by senescent and non-senescent cells in culture media, Human Cytokine Arrays (#ARY005; R&D Systems) were used, essentially following the manufacturer’s instructions. Briefly, individual conditioned media (CM) were first combined with the provided cocktail of biotinylated antibodies. The mixture was then incubated with the array membrane. Under these conditions, any cytokine present in the conditioned media would bind to a primary antibody immobilized on the membrane surface. Streptavidin–horseradish peroxidase and chemiluminescence detection reagents (ECL Prime Western Blotting, Amersham, GE healthcare, Chicago, IL, USA) were added, and the signal was detected upon exposure to an X-ray film (Fujifilm super HR-U, Allendale, NJ, USA). Densitometric analyses were then performed. The intensity of the signals was measured using image J software 1.45 (https://imagej.net/Downloads).

### 4.7. Adhesion Assays

In order to examine the ability of platelets to adhere to senescent cells in vitro, a layer of adherent senescent cells (1 μM Palbociclib, 72 h treatment) or DMSO-treated control cells was first generated on coverslips in 6-well plates. A 24 h post-treatment incubation in serum-free medium was then allowed. Following this period, 1 × 10^6^ of washed platelets, previously stained with calcein-AM (Santa Cruz Biotechnology), were directly added to senescent and non-senescent cells and incubated for 3 h under constant agitation. The cells were then extensively washed with 1× PBS, fixed in 4% paraformaldehyde/1× PBS pH 7.4. The nuclei of the cells were visualized with DAPI. Images of platelets were obtained with the green filter (FITC U-3N31001) and those of the cell nuclei with the blue filter (DAPI U-3N31000v2) using a BX53 fluorescence microscope (Olympus, Shinjuku, Tokyo, Japan) and the Capture Pro 7 software (QImagine, Inc., Surrey, British Columbia, Canada). The assays were performed in triplicate, and seven separate microscopic fields per replica were analyzed.

### 4.8. Cell Migration and Invasion Assays

Cell migration was assessed by wound-healing assays, essentially as described before [84]. Briefly, 1 × 10^6^ cells were seeded on a 60 mm-diameter dish and left to proliferate overnight in order to form a monolayer. Next day, the monolayer was scratched with a pipette tip, and the cells were allowed to migrate for 24 h in serum-free conditioned medium. The distance between the edges of each scratch was measured by microscopic inspection.

Invasion assays were performed using Transwell invasion chambers (Corning Inc., Corning, NY, USA) carrying membranes (6.5 mm diameter) with 8 μm-diameter pores. Each filter was previously treated with 10 μg/cm^2^ fibronectin. In total, 1 × 10^5^ non-senescent cells, resuspended in 200 μL of serum-free medium, were added to the upper chamber, whereas 500 μL of senescent cells-containing medium, with or without platelets, was added to the lower chamber. Following incubation for 6 h at 37 °C, cells that had traversed the membrane and reached its bottom surface were fixed with methanol and stained with crystal violet. Photographs were obtained, and the stained cells were counted by microscopic inspection. The assays were performed in triplicate, and seven separate microscopic fields per replica were analyzed.

### 4.9. Statistical Analyses

Data were compiled and analyzed with the Sigma Plot software package, version 12.0 (Systat Sofware, GmbH, Germany) Group differences were calculated with t-student or one-way ANOVA with post-hoc Tukey HSD test. Differences with *p* values < 0.05 were considered statistically significant, and all data were shown as mean ± standard error of the mean (SD).

## Figures and Tables

**Figure 1 ijms-20-05292-f001:**
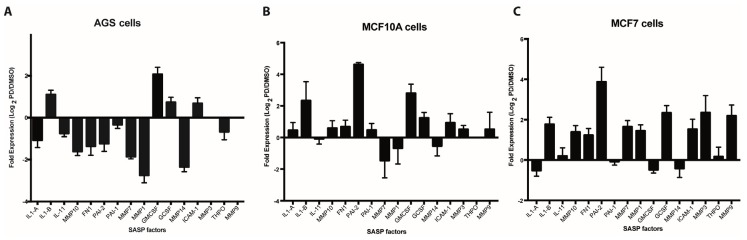
Expression analyses of senescence-associated secreted factors. Expression of selected senescence-associated secretory phenotype (SASP) factors (see Appendix A) by qRT-PCR in Palbociclib-induced senescent human gastric cancer cells (AGS) (**A**), immortalized mammary epithelial cells (MCF-10A) (**B**) and breast cancer cells MCF-7 (**C**) and their non-senescent DMSO-treated counterparts. Expression is expressed as log_2_ of the ratio between the expression values of Palbociclib-treated and DMSO-treated cells.

**Figure 2 ijms-20-05292-f002:**
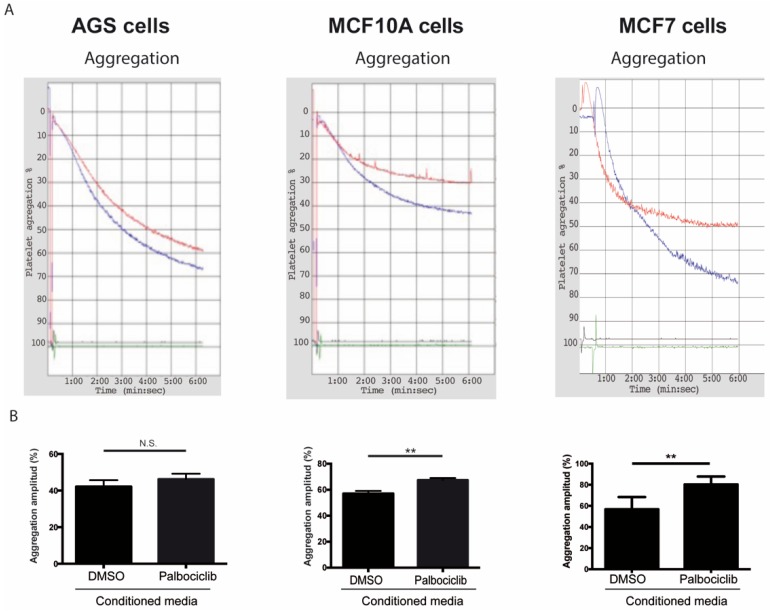
Effects of conditioned media from senescent cells on platelet aggregation. (**A**) Representative images of time-course recordings of platelet aggregation assays carried out in washed platelets incubated in conditioned media collected from senescent (blue curve) and non-senescent (red curve) AGS, MCF-10A, and MCF-7 cells. (**B**) The percentage of maximum platelet aggregation for three independent experiments was plotted (*n* = 3; ** *p* < 0.01; t-student test; N.S., not statistically significant).

**Figure 3 ijms-20-05292-f003:**
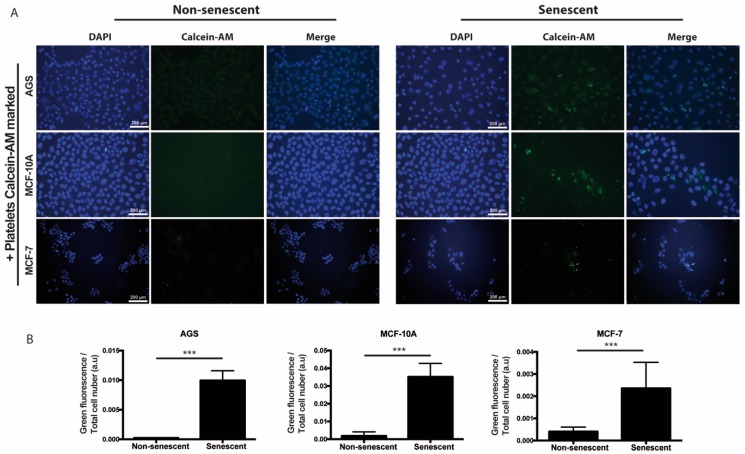
Adhesion of platelets to senescent cells in vitro. (**A**) Washed platelets, previously labeled with the fluorescent dye Calcein-M (green signal), were added to senescent or non-senescent cell cultures and further incubated for 1 h under standard culture conditions, before extensive washing, fixation, and labeling of the cell nuclei with DAPI (blue). (**B**) Quantification of green fluorescence (platelets) corrected by cell number is shown. The intensity of fluorescence was measured by using ImageJ software (*n* = 6; *** *p* < 0.001; t-student test; scale bars mean 200 μm).

**Figure 4 ijms-20-05292-f004:**
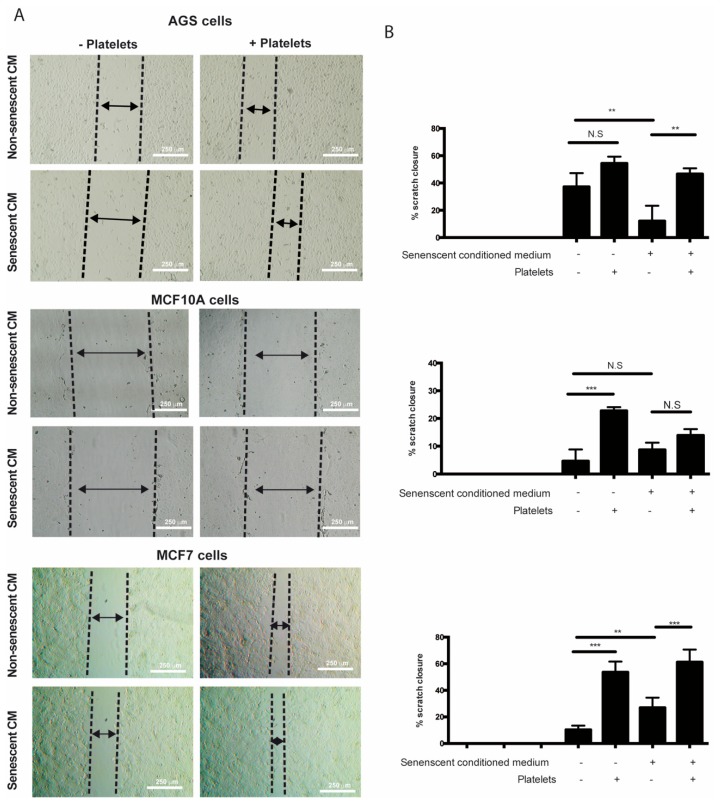
Effects of media conditioned by senescent cells and platelets on migration of non-senescent cells. (**A**) In vitro wound-healing assays were conducted in non-senescent AGS, MCF-10A, and MCF-7 cells exposed to conditioned media from senescent or non-senescent cells, with or without the presence of platelets, for 48 h. Migration distance from the edge of the wound (scratch) was measured at time 0 and following 48 h of exposure to conditioned media, platelets, or both. (**B**) Analyses of percentage of closure induced by senescent cells-conditioned media, platelets, or both; graphs represent the mean ± SD (*n* = 8; ** *p* < 0.01; *** *p* < 0.001, one-way ANOVA and Tukey’s multiple comparison post-hoc test; scale bars mean 250 μm; N.S., not statistically significant).

**Figure 5 ijms-20-05292-f005:**
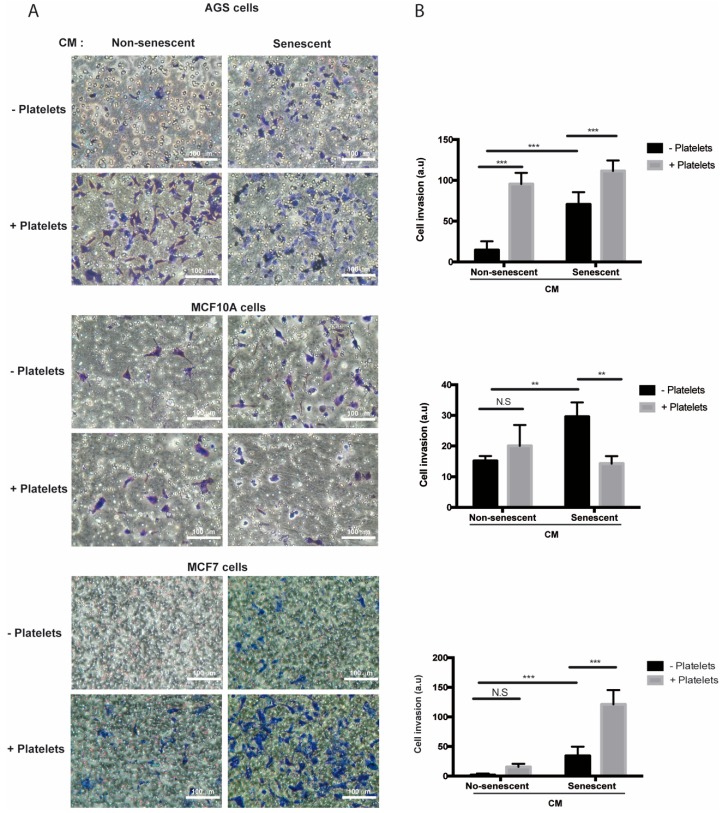
Effects of media conditioned by senescent cells and platelets on invasion of non-senescent cells. (**A**) Invasion assays were conducted in AGS, MCF-10A, and MCF-7 cells exposed to conditioned media from senescent or non-senescent cells, with or without the addition of platelets. Assays were carried out in fibronectin-coated Transwell chambers for 6 h. Cells that migrated to the lower face of the filter chamber were stained with crystal violet, photographed, and counted. (**B**) Analyses of cell invasion induced by senescent cells-conditioned media, platelets, or both; graphs represent the mean ± SD. (** *p* < 0.01; *** *p* < 0.001; *n* = 8 for each group, one-way ANOVA and Tukey’s multiple comparison post-hoc test; scale bars mean 100 μm; N.S., not statistically significant).

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
