# Peer review of "SASP-Dependent Interactions between Senescent Cells and Platelets Modulate Migration and Invasion of Cancer Cells"

_ijms, 2019, doi:10.3390/ijms20215292_

Round 1

Reviewer 1 Report

The authors have made considerable changes to the manuscript by the addition of data using a third cell line, the breast cancer cell line MCF-7. The message of the manuscript could be further clarified. The title ‘Factors secreted by senescent cells induce platelet aggregation’ would indicate that this is the main finding, yet the effect on aggregation is small. A much greater effect on platelet adherence and migration was observed.

The are a number of sentences that require to be clarified or there appears to be missing words e.g.

Line 43 – missing ‘of’ after formation. Line 80 – check sentence structure

Figure 2B MCF-7 cells – can the authors confirm that this result is significant, the difference does not look to be due to the size of the error bars. Can the authors confirm statistical significance on Figure 3B. No statistics are given. There are no scale bars on any of the microscopy images. Please add.

Reviewer 2 Report

This is a very interesting and well-performed study. I found some issues which need to be explained/corrected before publication, but I hope that will be no problem for the Authors.

Figure 4 and 5 need to be rearranged to have better visibility and quality

Reference number of cells, as well as detailed source of cell, should be provided

Can Authors provide some other characterisation of blood donors as well as how they were classified to tests?

Which type of anticoagulant was used? When were the blood samples collected?

Please provide the fibrinogen source

The language needs to be corrected I found a lot of grammatical errors

How Authors estimated the possibility of Tukey test use? How normality and equality of variance were verified?

Round 2

Reviewer 2 Report

All my comments have been included in revised version. I'm reccomending this paper for publication.

This manuscript is a resubmission of an earlier submission. The following is a list of the peer review reports and author responses from that submission.

Round 1

Reviewer 1 Report

The manuscript by Vaenzuela et al addresses an important question for age-related pathologies.  How does cellular senescence alter the reactivity of platelets?  The authors show an upregulation of senescence-associated secretory phenotype (SASP) factors is observed in MCF-10A cells and a minimal increase or downregulation of these factors in AGS cells.  Media from senescent MCF-10A cells enhanced platelet aggregation and adhesion to senescent cells and induced invasion.  AGS senescent cell media enhanced platelet adhesion to senescent cells but not platelet aggregation .  AGS senescent cell media also induced aggregation.  Platelets were also able to induced invasion in non-senescent cells treated with senescent cell media.  The paper has some interesting data and highlights the complex interaction of platelets with potential activation and chemotactic facors.

I have some points for consideration below;

1.     Which Serpin is overexpressed (Figure 1) in senescent MCF-10A cells?  The authors state that SERPIN, which I assume they are referring to SerpinB2 (PAI-2) as PAI-1 is labelled on the figure.  PAI-2 has a lesser role in fibrinolysis than PAI-1.  Perhaps more of significance here is the potential role of PAI-2 in metastasis.

2.     In AGS cells, gene expression of most SASP factors were downregulated or showed limited upregulation.  Can the authors comment on what other factors may be responsible for the effects on platelet adhesion and invasion.

3.     It is not clear from the text what the platelets were stimulated with to induce aggregation.  The methods suggest ADP and fibrinogen and mention a range of concentrations used.  Were these used as single or dual agonists?  The figure legend does not state the conditions used. 

4.     There are multiple spelling mistakes in figure 5.  The legend also does not state what the ‘a, b and c’ represent on the figure. 

5.     The authors state ‘platelets, alone or in combination with media conditioned by senescent MCF-10A cells, did not have any effects on invasion of non-senescent MCF-10A cells’.   Platelets appear to inhibit the induction of invasion induced by senescent CM.  The authors do not address this.  Can the authors comment on which factors may be involved in suppressing this response? 

6.     Targeting some of the upregulated SASP factors e.g. inhibiting IL-1β in migration/invasion assays would add valuable insight on a potential mechanism. 

Reviewer 2 Report

Blood platelets have been target of various studies which had tried to correlate their activation with cancer progression. This paper has very unique idea but I found to many issues to publish it in IJMS:

The introduction is too general and it is not showing the potential role of blood platelets in pathological states. Authors should more focused on platelets contribution in pathological conditions of human organism.

How in n=3 authors were able to obtain statistical significant p<0.001? This is impossible when they were use proper analysis. In all analysis SD should be presented not SEM which reduced the deviation between results. How Authors estimated the possibility to use Tukey Anova test? Why they did not use the non-parametric evaluation.

SERPIN is regulator of all proteolytic enzymes. Also these involved in pro-coagulant system why Authors did not mentioned about that. That is very important in role of thrombin in cancer progression.

Number of approve of the Human Ethics Committee must be provided.

The results obtained for platelet aggregation are very strange. In presented figures I do not see typical response for ADP (which was added according to description in materials and methods). So according to this methods authors checked the synergies between ADP and secreted molecules. Why authors did not checked possibilities to induce aggregation by pure secreted molecules? This will provide better results.

How Authors were able to determine the platelets by DAPI?, This dye is a fluorescent stain that binds strongly to adenine–thymine rich regions in DNA. But platelets are unnucleated cells without DNA!!!

Lines: 44, 53, 55, 58, 68 etc. wrong citation form.